# Green Financial Reform and Corporate ESG Performance in China: Empirical Evidence from the Green Financial Reform and Innovation Pilot Zone

**DOI:** 10.3390/ijerph192214981

**Published:** 2022-11-14

**Authors:** Zhao Chen, Ling Hu, Xin He, Ziming Liu, Danni Chen, Weirui Wang

**Affiliations:** 1School of Economics and Trade, Guangdong University of Foreign Studies, Guangzhou 510006, China; 2School of Finance, Jiangxi University of Finance and Economics, Nanchang 330013, China; 3School of Economics, Jinan University, Guangzhou 510630, China; 4Carey Business School, Johns Hopkins University, Baltimore, MD 21201, USA

**Keywords:** green financial reform, corporate ESG, green development, environmental mechanism, social mechanism and governance mechanism, green spillover

## Abstract

Does the establishment of pilot zones for green finance reform and innovations in 2017 have an impact on the Environment, Social and Governance (ESG) scores of enterprises? This paper selects data from Chinese A-share listed companies from 2014–2020 and uses the differences-in-differences (DID) model to analyze the impact of green financial reform on the ESG scores of enterprises. The study shows that the establishment of the Green Financial Reform and Innovation Pilot Zone (GFPZ) policy helps enterprises to obtain higher ESG scores through environmental, social and governance mechanisms. When ESG is measured using environmental, social and governance data, our results suggest that the contribution of green finance reforms to ESG scores is primarily driven by social responsibility scores. The adjustment effect analysis shows that for large enterprises in the GFPZ, the above effects have stronger influence, but there is no significant difference between heavily polluting and non-heavily polluting firms in the GFPZ. Expansive analysis shows that the improvement in ESG scores of enterprises in the GFPZ not only contributes to the green performance of enterprises, but also to their financial performance.

## 1. Introduction

In recent years, rapid economic development has brought about serious environmental problems, and China has become increasingly concerned about environmental issues, emphasizing the concept of sustainable and healthy development. Relevant national departments have also accelerated the rollout of green finance construction, emphasizing the sustainable development route of “Lucid waters and lush mountains are invaluable assets”. The focus is on the role of fiscal and financial policies to achieve decarbonization of the global economy, low carbon transition and climate resilient growth [1]. To deeply address environmental issues, China has vigorously promoted the exploration of environmental economic policies and introduced a series of green policies, including environmental tax, green credit, green bond and green insurance [2]. Environmental regulations and green credit reduce the availability of financing for polluting enterprises, and increase their financing constraints [3,4,5]. Green bonds are one of the important financing tools for companies to promote a green economy, but they are limited by local economic, institutional and environmental governance [6,7]. Green finance provides the appropriate funding for sustainable development, but in terms of the effects of the current implementation of the reform, there is no doubt that further improvements are still necessary [8].

To further accelerate the establishment of China’s green financial system as well as promote the sustainable development of green finance, in June 2017, the State Council made a decision to introduce green financial reform and innovation pilot zones with different emphasis and characteristics in parts of Zhejiang, Guangdong, Guizhou, Jiangxi and Xinjiang provinces. In 2019, Gansu was approved to join the pilot program. This pilot zone marks a new phase of “top-down” top-level design together with “bottom-up” regional exploration of green finance in China, which reflects a determination to implement green financial reform. 

The ultimate goal of listed enterprises is to maximize economic benefits, but enterprises should also actively fulfill their social responsibilities. Based on the stakeholder theory, if enterprises want to achieve their own survival and sustainable success, they must meet the demands of corporate stakeholders. According to existing studies, fulfilling corporate social responsibility actively is beneficial to improving financial performance [9,10] as well as corporate value [11,12,13]. Environment, Social and Governance (ESG) investment has gained widespread attention from all sectors, in pace with the increasing concern for green development and sustainability. The advantages of ESG will increase the value of a company and conversely, the disadvantages will decrease its value [14]. In the context of limited investor attention, investors pay attention to the ESG scores of enterprises and tend to give higher valuations to companies that are committed to sustainable development and environmental friendliness. In addition, high ESG scores are conducive to winning the trust of financial institutions such as banks allowing companies with these scores to obtain financing at lower costs. Therefore, companies are under pressure from stakeholders to be more socially aware and environmentally responsible, and they want companies to focus on socially responsible behavior that improves ESG scores to increase value for the company as well as for investors [15].

Although many studies have concluded that there are several issues that need to be addressed in the promotion of green finance reform in China, most scholars affirm the positive effects of the reform. The existing literature rarely links the establishment of green finance reform and innovation pilot zones with ESG investments of enterprises. This paper hopes to explore the impact of the Green Financial Reform and Innovation Pilot Zone (GFPZ) policy on the enterprise ESG score, and to affirm the positive role of this policy. The research goal of this paper is to use the differences-in-differences (DID) model as an empirical research tool to test the positive impact of the GFPZ policy on enterprise ESG score, and to further explore the moderating effects on the E, S as well as G investment subsets, with a view to verifying the implementation effect of China’s green finance policy. This will be achieved if it is found that the GFPZ policy can motivate enterprises to carry out green finance activities, improve their ESG investment scores and further improve their comprehensive competitiveness.

Based on the data of China’s A-share listed companies from 2014 to 2020, this paper takes the introduction of the Green Financial Reform and Innovation Pilot Zone (GFPZ) in 2017 as a quasi-natural test by using a double difference model to analyze the impact of green financial reform on enterprise ESG scores. The paper offers the following four main contributions. Firstly, we analyze the impact of green financial reform policies from the perspective of ESG scores. Our study complements existing research by showing that the introduction of the GFPZ policy helps firms in the pilot zones to obtain higher ESG scores. Secondly, we analyze by moderation effects that the above effects are more pronounced in large enterprises but do not differ significantly for heavily polluting enterprises. Thirdly, our results show that green finance reform positively affects ESG scores mainly by affecting social responsibility scores, but not firms’ environmental protection scores and corporate governance scores. Finally, our study finds that firms in the GFPZ can improve their ESG scores through environmental mechanisms, social mechanisms as well as governance mechanisms.

The remainder of the paper is designed as follows. Section 2 presents the literature review and research hypotheses. Section 3 describes the data and methodology of the study. Section 4 gives the empirical results, including basic regression, adjustment effect analysis and intermediary effect analysis. Section 5 explores the spillover effects of ESG performance. The last section is the conclusion of the paper.

## 2. Literature Review and Hypotheses

Green finance is emerging in the world in the context of sustainable development, and the world is developing responsible investment with the aim of integrating ESG factors into the investment decision-making process to achieve better risk control and sustainable investment returns [16]. Green finance is the solution for achieving a contract between the economy and nature [17] and it can drive a sustainable development path for Asian economies, a process that requires the financial sector to underpin the green transition [18]. China’s green finance has continuously developed in recent years and achieved remarkable results. In 2017, China decided to launch GFPZs in locations across five provinces (regions) to explore replicable and scalable experiences in institutional mechanisms and make new attempts to promote green finance. According to institutional theory, institutions have the function of constraining and influencing organizational behavior. In addition, as an institutional arrangement to support the transformation of the economy to a greener one, green finance is bound to have an impact on the behavior of micro enterprises. Therefore, many scholars have conducted further research on the impact of the GFPZ policy for enterprises, gaining insights for promoting the development of green finance in non-pilot regions in China in the future. Most scholars recognize the advantages of the China’s GFPZ policy and point out that it has greatly enhanced the green technology innovation capability and sustainable performance of pilot enterprises [19]. In summary, researches of green finance tend to focus on the economic consequences and environmental benefits of enterprises located in pilot zones, concluding that GFPZ policy has mainly positive impacts on enterprises; but there is less focus on the ESG score perspective of enterprises. With the introduction of the “double carbon” target and the popularization of the concept of green development, ESG scores of enterprises have become more and more important to investors and companies. Therefore, this paper concentrates on the impact of the GFPZ policy on ESG scores of enterprises, and provides new empirical evidence for enterprises to actively engage in green innovation activities together with improving their ESG performance.

China currently has several green policies, including the release of green guidelines and the promotion of the establishment of green financial reform and innovation pilot zones. The Green Credit Policy (GCP) published in 2012 can contribute to the development of green finance by influencing corporate investment and financing channels [20]. Furthermore, ESG investment was greatly promoted under the release of the “Guidelines for Establishing a Green Financial System” in 2016, a green policy that strongly promoted green investment [21]. Green investment has become a key driver in the energy sector, and its rapid growth is now mainly driven by China [22]. Governments need to inform, coordinate, incentivize and control activities related to green finance [23]. The year 2016 is known for the birth of green finance, and the introduction of the GFPZ policy is conducive to promoting the green technology innovation capacity of enterprises. Government can motivate the development of green manufacturing businesses in the form of loan guarantees and interest subsidies [24]. As an important green financial policy, the GFPZ policy can strongly promote the green investment of enterprises. Moreover, enterprises in the pilot zones have a stronger ability to innovate in green technology, which is conducive to their reducing instances of pollution at source, and promoting their environmental performance. This results in the ESG performance of enterprises in the pilot zones being stronger, with higher ESG scores than those of non-test zone enterprises. Therefore, we propose hypothesis 1 for the paper.

**Hypothesis** **1.**
*The establishment of the green finance reform and innovation pilot zone promotes the increase in ESG scores of enterprises in this pilot zone.*


The impact of the ESG scores of firms in the test area may be influenced by the size of the firm and the level of pollution in the industry. Based on this, this paper aims to investigate the moderating effects of firm size and industry pollution levels.

The scale effect of large enterprises facilitates access to important economic and strategic resources, including government subsidies. Moreover, large enterprises have better internal control and strategic decision-making, and can take a long-term view and focus on the sustainable development of the enterprise. ESG disclosure scores for large-cap companies are significantly higher than those for small and medium-sized ones because of their better governance capabilities [25]. Moreover, ESG scores are dependent on the resources being available to provide ESG data and the availability of the data itself. Since this is more readily available to large-scale companies, ESG investment performance is greater for these large-scale companies [26]. Small-scale enterprises, on the other hand, often find it difficult to survive including obtaining financial support such as bank credit, and are therefore more likely to face financing difficulties. A minority of research puts forward a different view, finding that small businesses are not necessarily less well organized than large companies in terms of their corporate social responsibility [27]. However, most academics recognize that big business is better for social responsibility. An increase in social responsibility means that companies are more willing to engage in projects such as corporate governance and environmental stewardship to improve their ESG scores. In the wake of the promulgation of the GFPZ policy, large enterprises in the pilot zone are more likely to receive investment and financing support from financial institutions due to their scale advantage and strong repayment ability, thus to face lower financing constraints, and to increase their green finance investment, allowing them to improve their ESG scores. Therefore, we present hypothesis H2-a:

**Hypothesis** **2-a.**
*Compared with small enterprises, the ESG score enhancement effect is greater for large enterprises in the test area.*


Companies in the heavily polluting industries pay huge environmental treatment costs for polluting the environment and damaging the ecology. The government disciplines heavily polluting businesses by imposing emissions taxes and environmental taxes on them [28,29]. Moreover, companies with poor environmental records are subject to stock market discounts [30]. In view of this, the heavily polluting enterprises will have a stronger incentive to reduce pollution through source treatment and end-of-pipe control, lowering the cost of environmental management, thereby improving their environmental performance and economic performance and driving up their ESG scores. The GFPZ policy can help push heavy polluters to fulfill environmental social responsibility and focus on green orientation, which reduces the cost of debt financing for heavy polluters [31]. This paper therefore hypothesizes that heavily polluting firms in the test area are more likely to be driven by this policy to increase green investment thus enhancing their core competitiveness with better ESG scores compared to firms in non-heavily polluting industries. Therefore, we obtain the hypothesis H2-b:

**Hypothesis** **2-b.**
*The ESG score enhancement effect is greater for heavily polluting firms in the test area compared to non-heavily polluting firms.*


## 3. Data and Methodology

### 3.1. Sample Selection and Data Sources

Because China’s GFPZ policy was launched in 2017, in order to explore its utility on corporate ESG scores, we select Chinese A-share listed companies from 2014 to 2020 as the original sample. According to the research needs, we screened the initial sample as follows: (1) exclude listed companies that have been delisted during the study period; (2) exclude non-arm’s length listed companies such as those marked ST or ST*; (3) exclude financial listed companies in consideration of the special characteristics of their financial statements; and (4) exclude samples with undisclosed financial data and incomplete key financial data. After the above screening, our study sample consists of 7414 observations from 1106 Chinese A-share listed companies over the period 2014–2020. To eliminate the influence of outliers, all control variables are winsorized at the 1% and 99% levels. CSR scores are from Bloomberg and the rest of the key financial data is from the CSMAR database.

The explanatory variable for this paper is *ESG*, which is measured using the logarithm of the Bloomberg ESG score. The model chosen for this paper is a double difference model to analyze whether the GFPZ policy in 2017 will have an impact on ESG scores of firms, hence the use of policy dummy variables and regional dummy variables in this paper. *Post* indicates a value of 1 if the firm is in 2017 and later years, and 0 otherwise. *Pilot* indicates that the firm is located in the province where the first batch of pilot zones were established in 2017 taking a value of 1, and 0 otherwise. The double difference variable is the cross-product term of the policy dummy variable and the region dummy variable, indicating the impact of changes in ESG scores of firms located in the pilot zone following the introduction of the 2017 GFPZ policy, and hence the direction and statistical significance of its estimated coefficients are of interest in this paper.

### 3.2. Model

We examine the impact of the China’s GFPZ policy on corporate ESG scores of firms by estimating the following model:$$ESG_{i,t}=\beta_{0}+\beta_{1}Post_{t}\times Pilot_{i}+\beta_{2}Post_{t}+\beta_{3}Pilot_{i}+\beta_{4}X_{i,t}+\delta_{i}+\lambda_{t}+\mu_{j}+\eta_{p}+\varepsilon_{i,t}$$
where $ESG_{i,t}$ describes the ESG score of company *i* in year *t*, $Post_{t}\;$ is a dummy indicating the years after 2017, while $Pilot_{i}$ is a dummy indicating the province where the first batch of green financial reform and innovation pilot zones were established in 2017. The term *X_i,t_* is the control variable, and includes *Age*, *Leverage*, *ROA*, *FCF*, *Q*, *Top10*, *Female*, *Certificate* and *Opinion*. Among them, *Age* is the logarithm of the number of years of establishment. *FCF* is the ratio of net cash from financing activities to total assets. Larger banks with more female board members perform better in terms of ESG disclosure, thus we choose *Female* as control variable, which describes the ratio of the number of female directors to the total number of directors. Certificate represents the number of environmental certification programs, and *ε_i,t_* is the disturbance term. In this paper, we control for the fixed effects relating to individual year, industry and province.

### 3.3. Descriptive Statistics

Table 1A shows the descriptive statistics of the main variables in this paper. The natural logarithm of the average ESG score of companies is about 3.04, the median is 3.03, and the median is slightly lower than the mean, indicating that the ESG scores of listed companies in China need to be further strengthened. The standard deviation of ESG scores in our sample is 0.31, indicating that the ESG performance of the sample firms varies widely. The mean of *Pilot* is 0.29, reflecting the sample in the pilot area being only 29%, indicating that the scope of domestic green finance pilot zones needs to be further expanded. In terms of gender, 10% of the chairmen in the sample are female. The median of *Certificate* is 0, indicating more than half of the sample companies did not obtain environmental certification projects. The mean of *Opinion* is 0.98, meaning the financial statements of most of the sample companies were of acceptable quality.

Table 1B presents the difference between mean and median of ESG in the pilot areas before and after the GFPZ policy. Both of them are significantly positive at the 1% level, tentatively verifying that ESG ratings changed between before and after the GFPZ policy. In the treated group, the mean of *ESG* is 3.084, which is higher than the mean of *ESG* in the control group, indicating China’s GFPZ policy has promoted the improvement of corporate ESG scores, thus preliminarily verifying the Hypothesis 1 in this paper.

## 4. Empirical Results

### 4.1. Time Trends

We have mapped the average change trend of enterprise ESG scores in the pilot area and non-pilot areas from 2014 to 2020 in Figure 1. Prior to the introduction of the GFPZ policy in 2017, the average change in trend of ESG scores of firms in the pilot area and non-pilot areas is similar, making the two graphs parallel. Since 2017, the ESG score in the pilot area has risen significantly higher than that in the non-pilot area, showing that the GFPZ policy has promoted the ESG scores for enterprises. Moreover, before 2017, the trends of environmental scores, social scores as well as corporate governance scores in the pilot area and non-pilot areas were basically the same, but after 2017, the increases in the pilot area are greater than those of the control group, with the social score of enterprises increasing the most, even exceeding the non-pilot area. This shows that the basic model has passed the parallel trend test and that China’s GFPZ policy has indeed improved the ESG scores of firms. 

### 4.2. DID Model Results

Following the Hausman test, we choose a fixed effects model for year, industry and province. Table 2 presents the benchmark regression results, reporting the impact of the GFPZ policy on the ESG scores of enterprises in the pilot area. The first three columns are the regression test results estimated by OLS, RE, and FE, respectively. Without considering other control variables, the coefficients of interaction term *Post***Pilot* are 0.021 and are significant at the 10% confidence level. The last three columns are the regression results after adding a series of control variables. The coefficients of interaction term are 0.023, and are all significant at the 5% confidence level, indicating that the release of the GFPZ policy does promote the increase of ESG scores for enterprises in the zone in question, and boost the development of local enterprises in three aspects: environment, society and corporate governance.

For the control variables, the coefficient of *Age* is significantly positive at the 1% level, meaning that the longer the life of a company, the better its business performance and thus the higher its ESG score. The coefficient of *Leverage* is significantly positive at the 1% level, which indicates that the higher the level of financial leverage of a company, the more conducive it is to improving its business management and obtaining more operating profits. In addition, the coefficient of *ROA* is significantly positive at the 1% level, which is because the stronger the profitability of the enterprise, the stronger its corporate governance, and accordingly, the higher its ESG score. The coefficient of third-party certification is significantly positive at the 1% level, which indicates that third-party certification will enhance stakeholders’ business trust in the enterprise, thus helping the enterprise to reduce financing constraints. Moreover, *Option* is significantly positive at the 1% level, meaning companies in the GFPZ with good audit profiles generally have lower risks, better business conditions and higher ESG scores.

### 4.3. Analysis of Moderating Effects

To verify whether the impact of China’s GFPZ policy on ESG scores differs between types of enterprises, this paper applies the moderating effect for analysis. We use total firm assets to measure firm size [32] and divide the sample into large and small firms according to their size in order to test the difference in the impact of the release of the GFPZ policy on the ESG scores for firms of different sizes. Moreover, we divide the sample into heavily polluting and non-heavily polluting enterprises according to the pollution situation in the region to test the differential impact of the establishment of the pilot zone on companies of different polluting levels.

According to Table 3A, it can be seen that by conducting *t*-tests for differences in means for the two grouping variables of large scale and high polluting degree, the *t*-values of all groups are significant at the 1% level, so grouping the sample by these two dimensions is beneficial for further research in this paper, exploring the inconsistent ESG score impacts of different types of firms.

Table 3B presents the regression results, the first two columns showing the impact of the release of the GFPZ policy on the ESG scores of large companies before and after the inclusion of the control variables. The second column shows the regression coefficient of the interaction term *Post***Pilot***BigSize* to be 0.032, significant at the 1% confidence level, controlling for other variables affecting ESG scores. Consistent with hypothesis H2-a, the impact of the establishment of the pilot zone on the fulfillment of corporate social responsibility is mainly reflected in large enterprises. The reason may be that large enterprises are mostly large-scale operations and can obtain more policy preferences in the pilot area. If they fulfill their social responsibility, this helps to broaden their financing and investment channels, and alleviate the problem of financing constraints. This allows them to gain more economic benefits and promote their rapid development, and therefore can significantly improve their ESG scores. By contrast, it is not easy for small-scale enterprises to obtain scale effect.

Columns 3 and 4 in Table 4 show the impact of the GFPZ policy on the ESG scores of the heavily polluting firms. The results show that with the consideration of time, industry and province fixed effects, the regression results are not significant with or without the inclusion of control variables. This violates hypothesis H2-b, indicating that there is no significant difference in the impact of the GFPZ policy on the ESG scores of companies with different levels of pollution. The reason for this may be that the sample in this paper is selected from 2014–2020. Over a longer period of time, the continuous implementation of the GFPZ policy may have a profound impact on companies in both heavy and low-pollution sectors, causing these companies to increase their social responsibility and improve their ESG scores.

### 4.4. Intermediary Mechanism Test

Based on the theoretical part of the analysis, this paper will further test through empirical analysis whether the China’s GFPZ policy promotes ESG scores through the mechanisms of the three different dimensions of environment (E), society (S) and governance (G).

(1)Environmental mechanism

For the first channel concerning the influence of environmental factors, this paper examines two indicators: corporate industrial pollution control investment and environmental management information disclosure. For corporate industrial pollution control investment (*DPI*), this paper selects the logarithm of the total industrial pollution control investment in different provinces of China. Meanwhile, corporate environmental management information disclosure (*ERS*) is measured by the logarithm of the total number of eight environmental information items in the CSMAR database, including the disclosure of environmental protection philosophy and objectives by enterprises. 

Observing the regression results in Table 4, the first three columns investigate whether the enterprises in the GFPZ can increase their ESG scores by increasing their industrial pollution control investments. Column 1 illustrates that firms in the GFPZ have positive ESG scores, which matches the baseline regression results. Column 2 illustrates that firms located in the GFPZ increase their industrial pollution control investment. Moreover, column 3 illustrates that firms in the GFPZ can boost their ESG scores by increasing their investment in industrial pollution treatment to promote environmental governance. The last two columns examine whether firms in the GFPZ can increase ESG scores by increasing the degree of environmental information disclosure. Column 4 illustrates that firms located in the GFPZ can significantly increase the degree of environmental information disclosure. Moreover, the results in column 5 illustrate that firms located in the GFPZ can reduce information asymmetry by increasing the level of environmental disclosure, attracting investor attention, and promoting environmental governance together with improving their ESG scores. Therefore, the above empirical findings show that firms in the pilot area are able to promote the effectiveness of corporate environmental governance through environmental mechanisms, thus increasing the their ESG scores.

(2)Social mechanism

For the second channel of influence on social factors, this paper focuses on two indicators: government-level policy subsidies and firm-level market capitalization. For government subsidies (*Subsidy*), this paper uses the logarithm of government subsidies as a measure. As for firm market capitalization (*MSV*), this paper uses the logarithm of firm market capitalization to measure it.

The first three columns of the regression results in Table 5 examine whether firms in the GFPZ can increase ESG scores by receiving more government subsidies. Column 2 illustrates that firms located in the GFPZ can receive more government subsidies due to government support. The results in column 3 illustrate that firms in the GFPZ can boost their ESG scores by receiving more government subsidies to promote governance at the social level. Finally, the last three columns investigate whether firms in the pilot zone can increase their ESG scores by increasing their market capitalization. Column 4 repeats the results of the benchmark regressions in this paper, column 5 shows that firms located in the pilot area positively contribute to the increase in firm market capitalization, and column 6 results demonstrate that firm market capitalization plays a fully mediating role between firms located in the GFPZ and ESG scores. 

In summary, companies in the GFPZ can promote the effectiveness of governance at the social level through social mechanisms including government subsidies and the market value of their company, which in turn increases its ESG score.

(3)Governance mechanism

For the third channel of influence on corporate governance factors, we examine both external financing constraints and internal corporate governance efficiency mechanisms. Under the bank-based financial system in China, bank loans are the most important external financing method. Therefore, this paper adopts the number of new bank branches in each province (*AHHI*) to represent the external financing constraint of enterprises, which is calculated by dividing the sum of the number of new private banks, new foreign banks, new urban and rural banks and new joint stock banks by 10,000. Moreover, since the executive internal pay gap affects the efficiency of corporate investment as well as corporate performance [33], we use the executive internal pay gap to measure the internal governance efficiency of the firm. For the executive internal pay gap (*SalaryGap*), this paper calculates the average compensation of the top three executives minus compensation per capita.

Looking at the regression results in Table 6, the first three columns examine whether firms in the test area can increase their ESG scores by moderating external financing constraints. Column 1 remains consistent with the results of the baseline regressions. Column 2 illustrates that firms located in the pilot area have more new bank branches nearby, which means these firms have more external financing channels and lower financing constraints. And column 3 illustrates that firms in the pilot area are able to alleviate financing constraints, promote corporate governance, as well as improve ESG scores by obtaining more external financing.

The last two columns investigate whether firms in the GFPZ can increase their ESG scores by improving the efficiency of internal corporate governance to ease internal financing constraints. Column 4 illustrates that firms located in the pilot area will be interested in enhancing the pay gap and thus increasing the internal governance efficiency of the firm. Finally, column 5 illustrates that firms located in the pilot zone can increase their ESG scores by alleviating internal financing constraints. Therefore, enterprises in the pilot area can promote the effectiveness of corporate governance through corporate mechanisms to alleviate the internal and external financing constraints, thus increasing the ESG scores of the enterprises.

## 5. Green Spillover Effects 

In this section, we analyze the impact of ESG scores obtained by firms in the GFPZ on their green performance as well as on their financial performance in order to analyze the spillover effects generated by their ESG scores. According to existing studies, corporate ESG performance can further increase the market valuation of a company through the moderating effect of green innovation [34]. ESG has a positive impact on corporate financial performance because the integration of ESG information helps corporate executives to make better decisions for each investment project [35,36]. We therefore tentatively conclude that the introduction of the GFPZ policy will be beneficial to the investment and financial performance of the business.

This paper uses *EID*, *GI* and *EG* to explore the influence of ESG scores on the green performance of enterprises. *EID* indicates the environmental disclosure status of enterprises, which is measured by the number of environmental disclosure items of 12 environmental liabilities and governance information disclosure items including wastewater emissions, COD emissions, and SO_2_ emissions. Higher *EID* indicates greater transparency of environmental information disclosure of enterprises, which is conducive to the establishment of an environmentally friendly social image of enterprises. *GI* represents the green innovation capability of enterprises. Since green invention patents can reflect the quality of green innovation of enterprises compared with green utility model patents and green design patents, we take the natural logarithm of the number of green inventions independently filed by enterprises plus one to measure the green innovation capability, the data is obtained from the GPRD database. In addition, *EG* represents an enterprise’s environmental management capability, measured by environmental management fees including sewage, cleaning, sewage and sanitation fees. Smaller *EG* represents lower corporate environmental management costs along with less environmental management pressure.

In addition, we further explore the impact of corporate ESG performance on financial performance using *FC*, *GBS* and *CRASH_t+_*_1_. *FC* refers to the financing constraint of a firm and is measured by the WW index, thus a larger value of *FC* indicates a greater financing constraint faced by enterprises, and vice versa. *GBS* is the logarithm of the size of green bond issuance, and the larger the *GBS*, the larger the size of green bond issuance as well as the stronger the ability to obtain green project funding. *CRASH_t+_*_1_ is a dummy variable for crash risk calculated based on market capitalization in circulation in the sub-market, taking a value of 1 if the stock is at risk of crashing and 0 otherwise. The smaller the crash risk side of the company, the more stable the company’s access to finance.

Table 7 shows the result of the spillover effects of ESG scores. The first three columns indicate the influence of the ESG score on the green performance for the enterprises in the GFPZ. The first column shows that the ESG coefficient is significantly positive at the 1% level, indicating that the increase in ESG scores of firms in the GFPZ improves the disclosure of environmental information and is helpful to reduce information asymmetry between firms and investors. The second column shows that the ESG scores of enterprises is positively related to their green technology innovation capability. A good ESG score in the GFPZ is conducive to improving the green technology innovation capability of companies and their green performance. The third column shows that the improvement in ESG scores of companies in the GFPZ can reduce environmental management costs, which reduces the additional costs associated with companies polluting the environment together with improving their environmental management capacity.

The regression results in the fourth to sixth columns indicate the influence of the ESG score on the financial performance of firms. The fourth column indicates that an increase in the ESG scores of enterprises in the GFPZ can reduce their financing constraints along with improving their financial performance. The results in the fifth column show that an increase in ESG scores of enterprises is conducive to an increase in the scale of green bonds issued by them. Companies with higher ESG scores have more spillover effects from green performance and can mitigate the financing challenges they face through green bond issuance. The last column suggests that an increase in the ESG scores of companies in the GFPZ makes them less vulnerable to the risk of share price collapse and maintains share price stability.

## 6. Robustness Tests

### 6.1. Placebo Test and Replacing Core Variables

Table 8 shows the results of the placebo test and ESG subscale scores. This paper enhances the robustness of the findings through a placebo test by advancing the establishment of the GFPZ by 2 years, defining 2013–2014 as the pre-event period and 2015–2016 as the post-event period, while testing the impact of this dummy policy on the ESG scores of the firms. The coefficients of the *FPost***Pilot* variables in the first two columns are not significant, indicating that there is no systematic difference in the change trend of the ESG score of the treated group and the control group excluding the impact of the GFPZ policy.

In addition, when the core variables ESG scores were replaced with three individual indicators E, S, and G scores, the regression coefficient of the variable *Post***Pilot* is 0.830 and significant at the 5% level when the explanatory variable is the *S* indicator; on the other hand, the coefficient is not significant when the explanatory variables were the *E* and *G* indicators, indicating that green financial reform enhances social responsibility of enterprises and affects the *S* scores of companies, but does not affect their initiatives or scores in environmental protection and corporate governance. In sum, the overall contribution of the GFPZ policy to ESG scores is mainly driven by social responsibility scores.

### 6.2. Adding Control Variables and PSM-DID

In addition to the existing control variables, regional and macro variables including financial output, SO_2_ emissions, GDP growth and PPI growth are added to maintain the robustness of the baseline regression, where *FD* is measured by dividing financial output by 100 and *SO_2_* is measured by dividing SO_2_ emissions by the logarithm of the value of 10,000.

The first three columns in Table 9 show the regression results after adding regional and macro variables, and the last two columns show the DID regression for the samples after 1:1 matching of the control group using PSM as a way to mitigate the endogeneity problem. This paper uses a logit model to perform nearest neighbor matching on two sets of samples. The regression coefficients for *Post***Pilot* are all significantly positive and this result keeps pace with the results of the benchmark regression.

## 7. Treatment of New Pilot Areas

In 2019, Lanzhou New District in Gansu was approved as a new pilot zone for green financial reform and innovation in China, thus changing the timing of events in the GFPZ from one to two. In order to exclude the possibility that the newly established pilot zone entrapped in the control group may endogenously affect the results of this paper, we therefore performed a multi-period DID test. In addition, Lanzhou New District is not the first batch of GFPZ and is the control group in this paper: however, it became the second batch of GFPZ in 2019, which may positively affect ESG investment scores. Therefore, placing Lanzhou New District in the control group of this paper may interfere with the regression results of this paper. Removal of the newly established pilot zone sample is beneficial to exclude a small level of interference from the new experimental area. 

The first two columns in Table 10 show the regression results of the multi-period DID. Column 2 indicates that after controlling for a series of dependent variables that affect the ESG factors of the firms, the *NPost***NPilot* coefficient is 0.023, which is significant at the 5% level, indicating a positive ESG score impact for the firms in Lanzhou New Area in Gansu as a new experimental zone. The latter two columns are the regression results for the experimental group excluding the firms in Gansu Lanzhou New Area. Column 4 indicates the effect of adding control variables, with a coefficient of 0.023 and significant at the 5% level for *Post***Pilot*, indicating that excluding the effect of the new area of the trial on the control group still does not alter the conclusions of the underlying regression from the previous section.

## 8. Conclusions

This paper discusses the relationship between the China’s GFPZ policy and ESG scores of firms. Taking the GFPZ policy in 2017 as an entry point, this paper empirically finds that the GFPZ policy helps firms receive more financial subsidies and alleviate financing constraints based on data from Chinese listed companies from 2014–2020, resulting in higher ESG scores for firms in this pilot region. Moreover, the paper finds that the overall contribution of green financial reform to ESG scores is mainly driven by social responsibility scores. 

The adjustment effect shows that the above-mentioned effects are more obvious in large enterprises, because their size is conducive to obtaining policy preferences for large-scale operations, broadening these enterprises’ investment and financing channels and improving their ESG scores. However, it is found that there is no significant difference between the above-mentioned effects and the pollution degree of enterprises. The reason for this result may be that the GFPZ policy was implemented in a strong and widespread manner, and had the effect of significantly improving ESG scores for both heavily polluting as well as non-heavily polluting firms. As for the intermediary mechanism analysis section, this paper verifies from the three pillars of ESG theory that GFPZ policy can alleviate the financing constraints of enterprises in the test area through three major channels—environmental, social and governance—by improving their investment attractiveness, enhancing their core competitiveness, and thus improving their ESG scores and ratings. In order to further explore the impact of the improvement of ESG scores of firms, this paper explores the performance premium of ESG scores. It is found that the improvement of ESG scores of enterprises in the pilot zone will produce a performance premium effect. On the one hand, it will bring about green performance such as increasing information disclosure, enhancing green technology innovation capacity and reducing environmental governance pressure. On the other hand, it will bring about enhanced financial performance such as easing corporate financing constraints, expanding access to green bond financing and reducing the risk of share price collapse.

With global warming and a growing sense of environmental responsibility among investors, ESG investment is gaining traction. This paper provides empirical evidence on the improvement of corporate ESG performance in the China’s GFPZ areas and highlights the importance of green finance reform in China. Reform is not a one-day process, but a process of continuous advancement and improvement. The power of the ‘visible hand’ of policy should be emphasized. Enterprises in a GFPZ can benefit from the green finance sector through policy incentives to improve their environmental, social and governance performance. At the same time, enterprises should also take the initiative and seize the opportunities brought about by green finance through the ‘invisible hand’ of the market, broaden their financing channels through green bonds, green insurance and other green financial sectors, increase investment in green technology and innovation, and enrich their green products in order to enhance their core competitiveness, in order to cope with the economic downturn brought about by the current new epidemic.

Our research examines whether China’s green finance reform and innovation pilot zone policy will have an impact on the ESG scores of enterprises, which may benefit future research on the impact of the GFPZ policy on the fulfillment of corporate social responsibility, especially in innovation. It is of guiding significance for researchers to analyze the effect of green finance policies on the eco-efficiency of enterprises, especially on the environmental investment and pollution regulation activities of enterprises.

## Figures and Tables

**Figure 1 ijerph-19-14981-f001:**
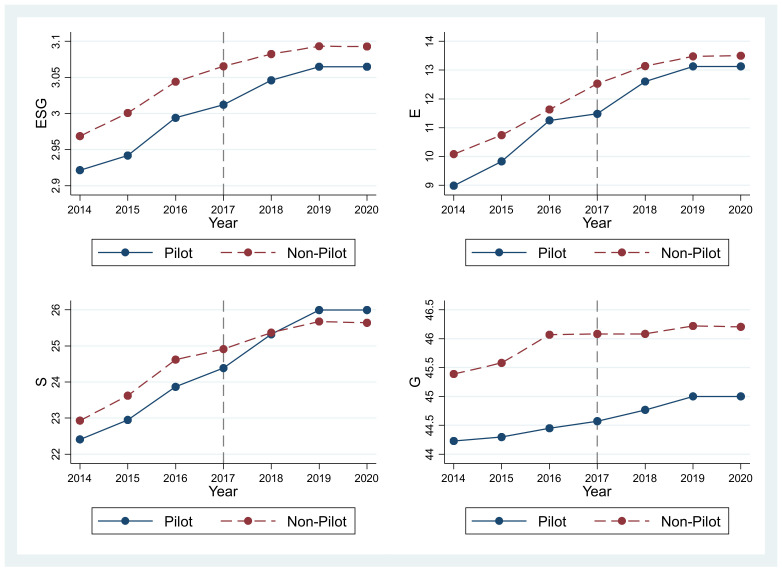
Time trend graph.

**Table 1 ijerph-19-14981-t001:** Summary statistics.

Panel A: Descriptive Statistics								
	Obs.		Max	Median		Min	Mean	Std. Dev.
*ESG*	7414		3.85	3.03		2.37	3.04	0.31
*Post*	7414		1.00	1.00		0.00	0.59	0.49
*Pilot*	7414		1.00	0.00		0.00	0.29	0.45
*Age*	7414		3.56	3.00		2.08	2.97	0.28
*Leverage*	7414		0.89	0.48		0.08	0.47	0.20
*ROA*	7414		0.24	0.04		−0.21	0.04	0.06
*FCF*	7414		0.32	−0.01		−0.21	0.01	0.09
*Q*	7414		9.69	1.67		0.85	2.19	1.54
*Top10*	7414		0.92	0.59		0.24	0.59	0.15
*Female*	7414		0.71	0.07		0.00	0.10	0.12
*Certificate*	7414		2.00	0.00		0.00	0.42	0.73
*Opinion*	7414		1.00	1.00		0.00	0.98	0.15
**Panel B: Difference between Mean and Median of ESG in the Pilot Areas Before and After the Pilot**								
**Variables**	**Treated Group**			**Control Group**			**MeanDiff**	**MedianDiff**
	**N**	**Mean**	**Median**	**N**	**Mean**	**Median**		
*ESG*	3097	3.084	3.067	1256	3.047	3.028	0.036 ***	6.395 ***

Note: This table reports the summary statistics of the variables in this paper. All control variables are winsorized at the 1% and 99% levels. *** indicates significance at the 1% levels.

**Table 2 ijerph-19-14981-t002:** DID model regression results of green finance reform on corporate ESG scores.

	(1)	(2)	(3)	(4)	(5)	(6)
	OLS	RE	FE	OLS	RE	FE
	*ESG*	*ESG*	*ESG*	*ESG*	*ESG*	*ESG*
*Post***Pilot*	0.021 *	0.021 *	0.021 *	0.023 **	0.023 **	0.023 **
	(1.83)	(1.83)	(1.84)	(2.07)	(2.07)	(2.11)
*Post*	0.133 ***	0.133 ***	0.133 ***	0.111 ***	0.111 ***	0.085 ***
	(17.02)	(17.02)	(17.09)	(8.38)	(8.38)	(2.69)
*Pilot*	−0.114	−0.114		−0.110	−0.110	
	(−1.57)	(−1.57)		(−1.58)	(−1.58)	
*Age*				0.108 ***	0.108 ***	0.186 *
				(3.07)	(3.07)	(1.88)
*Leverage*				0.126 ***	0.126 ***	0.082 ***
				(4.49)	(4.49)	(2.66)
*ROA*				0.130 ***	0.130 ***	0.103 **
				(3.02)	(3.02)	(2.36)
*FCF*				0.001	0.001	0.018
				(0.04)	(0.04)	(0.82)
*Q*				0.010 ***	0.010 ***	0.013 ***
				(4.11)	(4.11)	(5.39)
*Top10*				0.133 ***	0.133 ***	0.088 **
				(3.69)	(3.69)	(2.15)
*Female*				−0.025	−0.025	−0.013
				(−1.35)	(−1.35)	(−0.73)
*Certificate*				0.015 ***	0.015 ***	0.011 ***
				(5.13)	(5.13)	(3.59)
*Opinion*				0.023 *	0.023 *	0.020 *
				(1.89)	(1.89)	(1.66)
Constant	3.015 ***	3.015 ***	2.949 ***	2.528 ***	2.528 ***	2.275 ***
	(50.73)	(50.73)	(625.47)	(20.47)	(20.47)	(8.02)
Year Industry Province	YES	YES	YES	YES	YES	YES
Observations	7414	7414	7414	7414	7414	7414
Within R^2^	0.182	0.182	0.182	0.199	0.199	0.202

Notes: The *t*-values calculated under robust standard errors for firm-level clustering are in parentheses, and ***, ** and * indicate significance at the 1%, 5%, and 10% levels, respectively.

**Table 3 ijerph-19-14981-t003:** Empirical results of the moderating effects model.

Panel A: *t*-Test for Mean Difference in ESG Scores						
	Control Group		Experiment Group		Mean Difference	*t*-Value
	Obs.	Mean	Obs.	Mean		
*BigSize*	467	2.916	6947	3.047	−0.131	−8.927 ***
*HighPI*	5152	3.01	2262	3.104	−0.093	−12.140 ***
**Panel B: Regression Results of the Moderating Effects Model**						
			** *BigSize* **		** *HighPI* **	
			** *ESG* **	** *ESG* **	** *ESG* **	** *ESG* **
*Post***Pilot***BigSize*			0.045 **	0.061 ***		
			(2.12)	(2.73)		
*Post***Pilot***HighPI*					−0.002	−0.005
					(−0.10)	(−0.22)
*Post***Pilot*			−0.022	−0.034 *	0.021	0.025 *
			(−1.24)	(−1.79)	(1.55)	(1.85)
*Post***BigSize*			0.064 ***	0.047 ***		
			(4.08)	(3.00)		
*Post***HighPI*					0.001	0.001
					(0.12)	(0.07)
Constant			2.949 ***	2.263 ***	2.943 ***	2.268 ***
			(626.03)	(8.00)	(232.81)	(8.00)
Controls			NO	YES	NO	YES
Year Industry Province			YES	YES	YES	YES
Observations			7414	7414	7414	7414
Within R^2^			0.189	0.206	0.182	0.202

Notes: The t-values calculated under robust standard errors for firm-level clustering are in parentheses, and ***, ** and * indicate significance at the 1%, 5%, and 10% levels, respectively.

**Table 4 ijerph-19-14981-t004:** Environmental mechanism: *DPI* and *ERS*.

	Intermediate Variables: *DPI*			*ERS*	
	*ESG*	*DPI*	*ESG*	*ERS*	*ESG*
*Post***Pilot*	0.023 **	0.191 ***	0.019 *	0.132 *	0.022 **
	(2.11)	(7.89)	(1.83)	(1.72)	(1.99)
*DPI*			0.007 **		
			(2.29)		
*ERS*					0.012 ***
					(7.97)
*Post*	0.085 ***	−0.354 ***	0.089 ***	−1.968 ***	0.108 ***
	(2.69)	(−4.43)	(2.93)	(−10.07)	(3.45)
*Constant*	2.275 ***	6.523 ***	2.132 ***	−2.238	2.302 ***
	(8.02)	(7.90)	(6.76)	(−1.33)	(8.22)
Controls	YES	YES	YES	YES	YES
Year Industry Province	YES	YES	YES	YES	YES
Observations	7414	6310	6310	7414	7414
Within R^2^	0.202	0.166	0.199	0.378	0.215

Notes: The t-values calculated under robust standard errors for firm-level clustering are in parentheses, and ***, ** and * indicate significance at the 1%, 5%, and 10% levels, respectively.

**Table 5 ijerph-19-14981-t005:** Social mechanism: *Subsidy* and *MSV*.

	Intermediate Variables: *Subsidy*			*MSV*	
	*ESG*	*Subsidy*	*ESG*	*MSV*	*ESG*
*Post***Pilot*	0.023 **	0.201 ***	0.020 *	0.096 ***	0.016
	(2.11)	(3.86)	(1.87)	(2.82)	(1.49)
*Subsidy*			0.005 *		
			(1.67)		
*MSV*					0.060 ***
					(7.31)
*Post*	0.085 ***	0.099	0.085 ***	−0.065	0.085 ***
	(2.69)	(0.82)	(2.82)	(−0.69)	(2.87)
*Constant*	2.275 ***	−3.126 **	2.197 ***	−0.998	2.316 ***
	(8.02)	(−2.45)	(7.02)	(−1.14)	(8.60)
Controls	YES	YES	YES	YES	YES
Year Industry Province	YES	YES	YES	YES	YES
Observations	7414	6310	6310	7408	7408
Within R^2^	0.202	0.0906	0.199	0.459	0.226

Notes: The t-values calculated under robust standard errors for firm-level clustering are in parentheses, and ***, ** and * indicate significance at the 1%, 5%, and 10% levels, respectively.

**Table 6 ijerph-19-14981-t006:** Governance mechanism: *Subsidy* and *MSV*.

	Intermediate Variables: *AHHI*			*SalaryGap*	
	*ESG*	*AHHI*	*ESG*	*SalaryGap*	*ESG*
*Post***Pilot*	0.023 **	0.008 ***	0.020 *	0.001 *	0.022 **
	(2.11)	(10.41)	(1.86)	(1.79)	(1.99)
*AHHI*			0.395 *		
			(1.94)		
*SalaryGap*					1.171 ***
					(3.71)
*Post*	0.085 ***	−0.016 ***	0.091 ***	0.007 ***	0.074 **
	(2.69)	(−11.51)	(2.88)	(4.11)	(2.32)
*Constant*	2.275 ***	0.058 ***	2.252 ***	0.013	2.243 ***
	(8.02)	(4.66)	(7.95)	(0.87)	(7.77)
Controls	YES	YES	YES	YES	YES
Year Industry Province	YES	YES	YES	YES	YES
Observations	7414	7414	7414	7341	7341
Number of code	1106	1106	1106	1105	1105
Within R^2^	0.202	0.371	0.202	0.107	0.204

Notes: The t-values calculated under robust standard errors for firm-level clustering are in parentheses, and ***, ** and * indicate significance at the 1%, 5%, and 10% levels, respectively.

**Table 7 ijerph-19-14981-t007:** Spillover effects of ESG scores: green performance and financial performance.

	(1)	(2)	(3)	(4)	(5)	(6)
	*EID*	*GI*	*EG*	*FC*	*GBS*	*CRASH_t+_* _1_
*ESG*	1.279 ***	0.615 ***	−0.358 ***	−0.173 ***	0.022 *	−0.088 *
	(6.45)	(4.83)	(−3.06)	(−3.92)	(1.66)	(−1.70)
Constant	−5.401 **	−2.684 *	1.201	0.723	0.079	0.526
	(−2.37)	(−1.71)	(0.47)	(1.33)	(1.04)	(0.74)
Controls	YES	YES	YES	YES	YES	YES
Year Industry Province	YES	YES	YES	YES	YES	YES
Observations	7414	7414	4548	7414	7414	4548
Number of code	1106	1106	1102	1106	1106	1102
Year Industry Province	YES	YES	YES	YES	YES	YES
Within R^2^	0.346	0.290	0.0946	0.418	0.00530	0.00738

Notes: The t-values calculated under robust standard errors for firm-level clustering are in parentheses, and ***, ** and * indicate significance at the 1%, 5%, and 10% levels, respectively.

**Table 8 ijerph-19-14981-t008:** Robustness check: Placebo test and ESG subscale scores.

	Placebo Test		ESG Subscale Scores		
	*ESG*	*ESG*	*E*	*S*	*G*
*FPost***Pilot*	0.013	0.013			
	(1.10)	(1.10)			
*FPost*	0.135 ***	0.089 ***			
	(17.07)	(2.81)			
*Post***Pilot*			0.444	0.830 **	−0.002
			(0.98)	(2.05)	(−0.02)
*Post*			2.905 **	1.840 *	1.202 ***
			(2.14)	(1.71)	(2.84)
Constant	2.949 ***	2.287 ***	0.444	0.830 **	−0.002
	(625.59)	(8.04)	(0.98)	(2.05)	(−0.02)
Controls	YES	YES	YES	YES	YES
Year Industry Province	YES	YES	YES	YES	YES
Observations	7414	7414	6627	7395	7414
Within R^2^	0.181	0.200	0.156	0.0904	0.0424

Notes: The t-values calculated under robust standard errors for firm-level clustering are in parentheses, and ***, ** and * indicate significance at the 1%, 5%, and 10% levels, respectively.

**Table 9 ijerph-19-14981-t009:** Robustness check: adding control variables and PSM-DID.

	Adding Variables			PSM-DID	
	*ESG*	*ESG*	*ESG*	*ESG*	*ESG*
*Post***Pilot*	0.027 **	0.023 **	0.027 **	0.031 **	0.034 **
	(2.48)	(2.11)	(2.48)	(2.14)	(2.32)
*Post*	0.079 **	0.097 ***	0.080 **		−0.003
	(2.21)	(3.39)	(2.51)		(−0.05)
*FD*	−0.445		−0.445		0.586
	(−0.99)		(−0.99)		(0.63)
*SO_2_*	−0.011		−0.011		−0.024
	(−1.55)		(−1.55)		(−1.29)
*GDP*		0.236 *	0.014		0.400
		(1.94)	(0.07)		(0.93)
*PPI*		−0.100	−0.047		−0.088
		(−0.71)	(−0.32)		(−0.29)
Constant	2.257 ***	2.255 ***	2.255 ***	2.927 ***	1.537 ***
	(7.65)	(7.87)	(7.55)	(357.58)	(2.59)
Controls	YES	YES	YES	YES	YES
Year Industry Province	YES	YES	YES	YES	YES
Observations	6941	7414	6941	2742	2742
Within R^2^	0.206	0.202	0.206	0.186	0.205

Notes: The t-values calculated under robust standard errors for firm-level clustering are in parentheses, and ***, ** and * indicate significance at the 1%, 5%, and 10% levels, respectively.

**Table 10 ijerph-19-14981-t010:** Robustness check: multiple-stage DID and deleting samples.

	Multi-Phase DID		Deleting Samples	
	*ESG*	*ESG*	*ESG*	*ESG*
*NPost***NPilot*	0.020 *	0.023 **		
	(1.83)	(2.11)		
*Post***Pilot*			0.021 *	0.023 **
			(1.82)	(2.09)
*Post*			0.133 ***	0.086 ***
			(17.01)	(2.73)
*Constant*	2.949 ***	2.275 ***	2.949 ***	2.285 ***
	(625.47)	(8.02)	(622.32)	(8.05)
Controls	NO	YES	NO	YES
Year Industry Province	YES	YES	YES	YES
Observations	7414	7414	7367	7367
Within R^2^	0.182	0.202	0.183	0.202

Notes: The t-values calculated under robust standard errors for firm-level clustering are in parentheses, and ***, ** and * indicate significance at the 1%, 5%, and 10% levels, respectively.

## Data Availability

The datasets used and/or analyzed during the current study are available from the correspond-ing author on reasonable request.

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
