# Peer review of "Green Financial Reform and Corporate ESG Performance in China: Empirical Evidence from the Green Financial Reform and Innovation Pilot Zone"

_ijerph, 2022, doi:10.3390/ijerph192214981_

Round 1

Reviewer 1 Report

This topic sounds interesting and can be a potential for publication. The issues for improvement are provided. Before resubmitting, please consider the following.

1.     What are your research questions (RQs) and research objectives (ROs)? The authors need to develop RQs/ROs and explain why the study is needed and what are the justifications for undertaking this study.

2.     The authors need to identify gap(s) of the existing studies and propose this study (see No. 2).

3.     In Line 51, please give the full name of ‘ESG’ when using at the first time.

4.     In Line 64, please give the full name of ‘DID’ when using at the first time.

5.     In Line 67, please give the full name of ‘GFPZ’ when using at the first time.

6.     In Section 3, there is a lack of data analysis. How did the authors analyze the data?

7.     In Section 6.4, can the authors explain why the findings did not alter the underlying regression from the previous section?

8.     Line 508 should be Section 7.

9.     The authors need to add managerial implications explaining how stakeholders can benefit from the study. In addition, the authors need to add contributions to theory and literature explaining how this research fills or advances the existing theories.

10.  Can the authors suggest some recommendation to further research?

11.  There are a mix of US and UK English style such as ‘analyze’ and ‘analyse’. Please re-check the consistency of the whole manuscript.

12.  Please re-check the journal requirement for a manuscript structure such as reference format.

Overall, you have good research. Following the above suggestions would overcome deficiencies in your manuscript.

Author Response

Dear Reviewer:

Thank you for your comments concerning our manuscript entitled “Green Financial Reform and Corporate ESG Performance in China: Empirical Evidence from the Green Financial Reform and Innovation Pilot Zone” (ijerph-1983717). Those comments are all valuable and very helpful for revising and improving our paper. We have studied comments carefully and have made corrections which we hope meet the approval. The main corrections in the paper and the responses to your comments are following:

Point 1: What are your research questions (RQs) and research objectives (ROs)? The authors need to develop RQs/ROs and explain why the study is needed and what are the justifications for undertaking this study

Response 1: Your comments are really thoughtful. we have added the RQs and ROs in the introduction. We hope to explore the impact of the Green Financial Reform and Innovation Pilot Zone (GFPZ) policy on the enterprise ESG score and affirm the positive role of the GFPZ policy. The research goal of this paper is to use the Differences-in-Differences (DID) model, empirical research tests the positive impact of GFPZ policy on enterprise ESG score and further explores the moderating effects and the effects on the E, S as well as G investment subsets, verifying the implementation effect of China's green finance policy. Therefore, it is found that the GFPZ policy can motivate enterprises to carry out green finance activities, improve their ESG investment scores and further improve their comprehensive competitiveness.

Point 2: The authors need to identify gap(s) of the existing studies and propose this study.

Response 2: According to your comments, we have added the gaps of the existing studies and propose this study in Part 1 and Part 2. The existing literature rarely links the establishment of green finance reform and innovation pilot zones with ESG investments of enterprises. This paper would like to further focus on the impact of the GFPZ policy on ESG scores of enterprises, and provide new empirical evidence for enterprises to actively engage in green innovation activities together with improving their ESG performance.

Point 3: In Line 51, please give the full name of ESG' when using at the first time.

Response 3: Your comments are really helpful. We have given the full name of ESG' when using at the first time.

Point 4: In Line 64, please give the full name of DID' when using at the first time.

Response 4: Your comments are really helpful. We have given the full name of DID' when using at the first time.

Point 5: In Line 67, please give the full name of GFPZ' when using at the first time

Response 5: Your comments are really helpful. We have given the full name of GFPZ' when using at the first time.

Point 6: In Section 3, there is a lack of data analysis. How did the authors analyze the data?

Response 6: We appreciate it very much for this good suggestion, and we have done it according to your ideas. We have supplemented the content of the data analysis in Section 3.

Point 7: In Section 6.4, can the authors explain why the findings did not alter the underlying regression from the previous section?

Response 7: Your comments are really helpful. We have explained why the findings did not alter the underlying regression from the previous section. Lanzhou New District is not the first batch of GFPZ and is the control group in this paper, but it became the second batch of GFPZ in 2019, which may positively affect ESG investment scores. Therefore, placing Lanzhou New District in the control group of this paper may interfere with the regression results of this paper. Removal of the newly established pilot zone sample is beneficial to exclude a small number of interference from the new experimental area.

Point 8: Line 508 should be Section 7.

Response 8: We have done it according to your ideas.

Point 9: The authors need to add managerial implications explaining how stakeholders can benefit from the study. ln addition, the authors need to add contributions to theory and literature explaining how this research fills or advances the existing theories.

Response 9: Your comments are really helpful. We have added the managerial implications explaining how stakeholders can benefit from the study in Part 1. In the case of limited investor attention, investors pay attention to the ESG score of enterprises and tend to give higher valuations to companies that are committed to sustainable development and environmental friendliness. In addition, companies with high ESG scores are conducive to winning the trust of financial institutions such as banks and obtaining lower financing costs. Therefore, companies are under pressure from stakeholders to be more socially aware and environmentally responsible, and they want companies to focus on socially responsible behavior that improves ESG scores to increase value for the company as well as investors.

Point 10: Can the authors suggest some recommendation to further research?

Response 10: We have added some recommendation to further research at the end of the paper. We examine whether China's green finance reform and innovation pilot zone policy will have an impact on the ESG score of enterprises. For future research, researchers can further explore the impact of the GFPZ policy on the fulfillment of corporate social responsibility, especially in green innovation. Researchers may analyze the effect of green finance policies on the eco-efficiency of enterprises, especially on the environmental Investment and pollution regulation activities of enterprises.

Point 11: There are a mix of US and UK English style such as 'analyze' and 'analyse' Please recheck the consistency of the whole manuscript.

Response 11: We are very sorry for our incorrect writing. We have examined these questions carefully and corrected them in the manuscript.

Point 12: Please re-check the journal requirement for a manuscript structure such as reference format.

Response 12: We have re-checked the journal requirement for a manuscript structure.

Reviewer 2 Report

Paper adequatly reffers to main variables of green finances in practices and theory of economics. Furthermore, authors have done a highly advisable and important inclusion of social, governmental and cultural variables of Corporate ESG. Paper therefore, as the wide paradigm of CSR may be due to green finances, includes  holistic framework crucial when evaluating green finance impacts in theory and practice.

Advice for future research to authors: Determination of CSR social impacts done by green finances, to declare how efficient they are (eco-efficency), especiall in terms of innovations.

Author Response

Dear Reviewer:

Thank you for your comments concerning our manuscript entitled “Green Financial Reform and Corporate ESG Performance in China: Empirical Evidence from the Green Financial Reform and Innovation Pilot Zone” (ijerph-1983717). Those comments are all valuable and very helpful for revising and improving our paper. We have studied comments carefully and have made corrections which we hope meet the approval. The main corrections in the paper and the responses to your comments are following:

Point 1: Advice for future research to authors: Determination of CSR social impacts done by green finances, to declare how efficient they are (eco-efficency), especially in terms of innovations.

Response 1: Your comments are really thoughtful. We have added some recommendation to further research at the end of the paper. We examine whether China's green finance reform and innovation pilot zone policy will have an impact on the ESG score of enterprises. For future research, re-searchers can further explore the impact of the GFPZ policy on the fulfillment of corporate social responsibility, especially in innovation. Researchers may analyze the effect of green finance policies on the eco-efficiency of enterprises, especially on the environmental Investment and pollution regulation activities of enterprises.

Reviewer 3 Report

In my opinion, the article should be supplemented with a conformation of the obtained results with previous research results.

Author Response

Thank you for your comments concerning our manuscript entitled “Green Financial Reform and Corporate ESG Performance in China: Empirical Evidence from the Green Financial Reform and Innovation Pilot Zone” (ijerph-1983717). Those comments are all valuable and very helpful for revising and improving our paper. We have studied comments carefully and have made corrections which we hope meet the approval. The main corrections in the paper and the responses to your comments are following:

Point 1: In my opinion, the article should be supplemented with a conformation of the obtained results with previous research results.

Response 1: We appreciate it very much for this good suggestion, and we have done it according to your ideas. We have pointed out this part in the introduction and literature review. Like most studies, this paper affirms the incentive effect of green finance reform and innovation pilot zone on enterprises, and points out the innovation points of this paper. This paper takes ESG score as the entry point. The existing literature rarely links the establishment of green finance reform and innovation pilot zones with ESG investments of enterprises. This paper would like to further focus on the impact of the GFPZ policy on ESG scores of enterprises, and provide new empirical evidence for enterprises to actively engage in green innovation activities together with improving their ESG performance.